# Mixture of Pre-processing Experts Model for Noise Robust Deep Learning on Resource Constrained Platforms

## Abstract

Deep learning on an edge device requires energy efficient operation due to ever diminishing power budget. Intentional low quality data during the data acquisition for longer battery life, and natural noise from the low cost sensor degrade the quality of target output which hinders adoption of deep learning on an edge device. To overcome these problems, we propose simple yet efficient mixture of pre-processing experts (MoPE) model to handle various image distortions including low resolution and noisy images. We also propose to use adversarially trained auto encoder as a pre-processing expert for the noisy images. We evaluate our proposed method for various machine learning tasks including object detection on MS-COCO 2014 dataset, multiple object tracking problem on MOT-Challenge dataset, and human activity classification on UCF 101 dataset. Experimental results show that the proposed method achieves better detection, tracking and activity classification accuracies under noise without sacrificing accuracies for the clean images. The overheads of our proposed MoPE are 0.67% and 0.17% in terms of memory and computation compared to the baseline object detection network.

## 1 Introduction

Internet-of-Things (IoT) platforms with distributed edge devices promise tremendous enhancements in many application domains including smart-cities, traffic management, surveillance, security, energy, health-care, and defense, to name a few. In a smart IoT platform, an edge device captures the data from sensors and processes it to extract meaningful information for pre-defined tasks. Deep learning promises major advancements in the quality of information that can be automatically extracted from streaming sensor data. The deep neural networks (DNN) can be deployed at the edge device and/or at the cloud to extract information.

Data captured by real-time sensors can be distorted during acquisition, in particular, when hardware operates under power constraints. For example, the spatial resolution of an image sensor can be reduced on-demand to save power, but low-resolution distorts the captured image. Likewise, low-power operation can enhance random noises introduced by a sensor. Therefore, a critical challenge for deploying DNN in smart IoT platforms is to meet the target output quality for a given task even with distorted and noisy input.

Data augmentation and denoising techniques can be used to make a DNN robust against input perturbations, however, it lowers detection accuracy for the clean images due to the regularization effect for the noisy images (Na et al., 2018). Mixture of experts can be used with the gating network which discriminates input distributions and multiple object detection networks trained with images from different distributions (Jacobs et al., 1991; Jordan & Jacobs, 1994). The model requires large memory for each expert, thus, it is impractical to use on the edge devices. On the other hand, lightweight average filtering, commonly used for image noise reduction, introduces undesired spatial distortions that can negatively impact accuracy for 'good' images, specially, for complex tasks such as object detection or temporal activity classifications.

This paper presents a lightweight mixture of pre-processing experts (MoPE) model for noise robust deep learning. We use the pre-processing to minimize the variance of the input images for the object detection network. Adversarially trained auto-encoder with skip connection is used as a

Table 1: Mean Average Precision @ IoU=0.5 on MS-COCO 2014 validation dataset. Faster R-CNN architecture (Ren et al., 2015) with inception v2 (Szegedy et al., 2016) as a backbone network is used. $\sigma$ : standard deviation for Gaussian noise when the image values are in [0,1]. $max\_\sigma = 0.15$ is used during training. As shown in this table, similarity loss is not effective for the object detection task.

| | mAP @ IoU=0.5 | | |
| Training | Similarity loss | Clean | $\sigma = 0.15$ |
| --- | --- | --- | --- |
| Clean+Noisy | - | 0.458 | 0.305 |
| Clean+Noisy | ✓ | 0.457 | 0.306 |

pre-processing unit for the noisy images and identity unit is used as a pre-processing for the clean and low resolution images. Gating network is trained to learn input distribution and used to select proper pre-processing units given input distributions.

The proposed approach is applied to DNNs designed for different tasks including object detection, multiple object tracking, and human activity classification. Experimental results demonstrate following key contributions:

- For distorted images/videos, our approach achieves better detection accuracy than a baseline network trained with only data augmentation.

- For clean images/videos, our approach demonstrates accuracy comparable to a vanilla network, and better than a network attached just with pre-processing modules (without MoPE).

- The proposed MoPE approach is effective for different classes of DNN including convolutional and recurrent networks.

- The complexity analysis shows the proposed approach adds negligible overheads (0.67% and 0.17% in terms of memory and computation) to the baseline DNNs.

## 2 RELATED WORK

**Denoising techniques:** Several image denoising algorithms exist in the literature (Buades et al., 2005; Dabov et al., 2007; Tomasi & Manduchi, 1998). Those algorithms assumed data distribution as a known prior, and proposed fine-tuned algorithms for the specific noise distributions. Milani et al. (2012) proposed a much simpler adaptive filtering strategy for Histogram of Oriented Gradients (HOG) based object detection in noisy images. Recently, convolutional neural network (CNN) based object detection with noisy images is proposed in Milyaev & Laptev (2017). They used an adaptive bilateral filter that considers local texture properties to decide the amount of intensity smoothing required. They also showed the ability of CNN to adapt with image noise by re-training it with noisy images. However, they have not reported how this re-trained network performs with clean images with or without pre-processing. We observed that applying pre-processing on clean images degrades the accuracy of re-trained network.

**Training techniques:** Training with data augmentation can always be used to increase the robustness of the network. Applying similarity loss (Na et al., 2018) on top of that has shown to increase noise robustness for image classification task. However, we show that it is not beneficial for the complex task like object detection which we would like to solve. To compare the effect of similarity loss for the object detection network, two faster R-CNN networks (Ren et al., 2015) have been trained with and without similarity loss. We have applied similarity loss to the output of the first feature extractor in the Faster R-CNN. Table 1 shows that the difference between mAPs for those networks is negligible. The results differ from the results for the image classification task as in Na et al. (2018). Possible explanation would be that the similarity loss is effective only for task which deals with continuous variables. Even though region proposal network uses the same features with the image classifier, generation of bounding boxes relates with many discrete components including anchors and number of region proposals, etc. And the object detection accuracy is determined based on the results from both region proposal network and image classifier at the second stage.

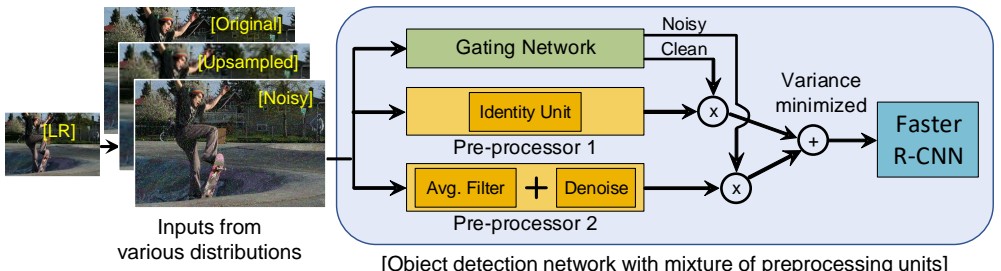

Figure 1: Object detection with mixture of pre-processing experts model

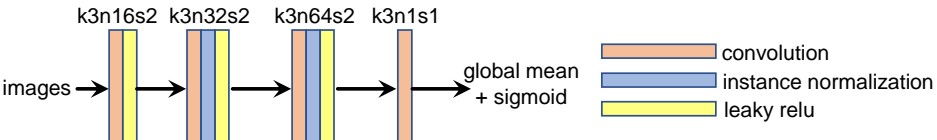

Figure 2: Gating network architecture

## 3 MIXTURE OF PRE-PROCESSING EXPERTS

Reasonable amount of image distortion is useful as a regularization when the test data distribution is expected to be in the same distribution with the training data. In our scenario, the input distribution is diverse including various resolution and noisy images. Thus, data augmentation increases the overall detection accuracies, however, it reduces individual accuracy compared to the network trained only on the specific data distribution. Table 2 shows that the network trained with clean, low resolution and noisy images improves detection accuracy for low resolution and noisy images compared to the baseline at the expense of decreased accuracy.

To tackle this problem, we propose simple MoPE model to incorporate various input distributions as shown in Figure 1. Each pre-processing expert is trained on the target input distribution like low resolution or noisy images and used as a pre-processing for the object detection network. Pre-processing unit trained on each data distribution minimizes the variance of the input distribution seen by the object detection network. A gating network is used to discriminate the input distribution whether the image is clean or noisy as shown in Figure 1. The gating network selects an input image for the object detection network among the output candidate images from pre-processing units.

### 3.1 GATING NETWORK

We use modified PatchGAN (Isola et al., 2017) for the gating network. We use 31x31 receptive filed, which is smaller than that from the original paper as Gaussian noise in nature is easy to identify in a relatively small region. As shown in Figure 2, we use 3x3 kernels for convolution layers, instance normalization (Ulyanov et al., 2016) and use shallower network than those in the original paper. It is also well aligned with our objective which is to make minimal changes for the edge devices, and it will be discussed in Section 6.4.

### 3.2 PRE-PROCESSING UNITS

**Pre-processing for the clean and low-resolution images:** We use no pre-processing for the high-resolution images and up-sampled version of low resolution images. As the objective of using the mixture of the pre-processing units was to develop a computationally efficient architecture, we found data augmentation was sufficient to enhance the robustness against high-resolution and low-resolution images.

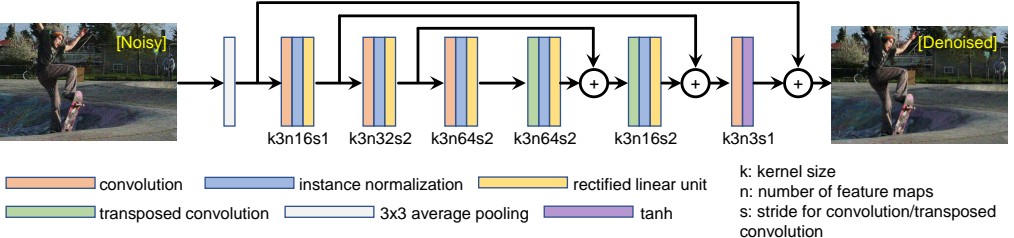

Figure 3: Denoise network architecture

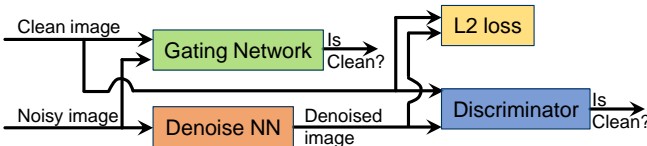

Figure 4: Training framework for denoise network with similarity and adversarial loss

**Pre-processing for the noisy images:** We consider average filter and denoise neural network as a candidate pre-processing for the noisy images. Average filter was very powerful pre-processor for the noisy images considering its simpleness, thus, included in our experiments. We also consider U-Net like denoise neural network (Ronneberger et al., 2015) as shown in Figure 3. We intentionally use the simple architecture to reduce the computational complexity. All the convolution and transposed convolution have 3x3 kernels, and the number of filters increases/decreases by 2 when the features are down/up sampled with stride 2. As our denoise network is pixel to pixel transformation between similar images, skip connection helps preserve the color information in original images, thus, eventually produce better quality. Average filter at front reduces the random noise in images at the expense of loss the edge information. Loss of the edge information is recovered by training denoise network with adversarial loss (Goodfellow et al., 2014) which will be discussed in the following section.

## 4 TRAINING OBJECTIVES

### 4.1 DENOISE NETWORK

We use adversarial loss (Goodfellow et al., 2014) on top of the similarity loss (L2 loss) as shown in Figure 4 for realistic denoising. For the noise function $F : X \rightarrow Y$ and denoise function $G : Y \rightarrow X$ and its discriminator $D$, we express the objective as:

$$\mathcal{L}_{\text{GAN}}(G, D|F) = \mathbb{E}_{x \sim p_{\text{data}}(x)}[\log D(x)] + \mathbb{E}_{y \sim p_{\text{data}}(y)}[\log(1 - D(G(y)))], \tag{1}$$

where $G$ aims to generate images $G(y)$ that look similar to images from clean data distribution $X$, while $D$ tries to distinguish between denoised images $G(y)$ and the original images $x$ for a given noise function $y = F(x)$.

We add similarity loss that encourages the denoised images $G(F(x))$ to be similar to the original images $x$:

$$\mathcal{L}_{\text{sim}}(G|F) = ||G(F(x)) - x||_2^2. \tag{2}$$

The total objective is:

$$\mathcal{L}(G, D|F) = \mathcal{L}_{\text{GAN}}(G, D|F) + \lambda \mathcal{L}_{\text{sim}}(G|F), \tag{3}$$

where $\lambda$ controls the relative importance of the two objectives. We use $\lambda = 1$ in the following experiments. Finally, we solve:

$$G^* = \arg \min_G \max_D \mathcal{L}(G, D|F). \tag{4}$$

Table 2: Mean Average Precision @ IoU=0.5 on MS-COCO 2014 validation dataset. Faster R-CNN architecture (Ren et al., 2015) with inception v2 (Szegedy et al., 2016) as a backbone network is used. LR: low resolution, for training we augment 2x and 4x down-sampled versions and use 4x down-sampled images for evaluation. $\sigma$ : standard deviation for Gaussian noise when the image values are in [0,1]. $max\_\sigma$ = 0.15 is used during training.

| mAP @ IoU=0.5 | | | | | | | |
|---|---|---|---|---|---|---|---|
| Model | Training | Pre-processing | Gating | Fine-tune | Clean | LR | $\sigma$ = 0.15 |
| 1 | Clean only | - | - | - | **0.461** | 0.335 | 0.226 |
| 2 | Clean only | Average filter | - | - | 0.445 | 0.336 | 0.299 |
| 3 | Clean only | Average filter | ✓ | - | 0.457 | 0.335 | 0.298 |
| 4 | Clean only | Denoise | - | - | 0.449 | 0.335 | 0.324 |
| 5 | Clean only | Denoise | ✓ | - | 0.457 | 0.335 | 0.323 |
| 6 | Clean+LR+Noisy | - | - | - | 0.454 | 0.360 | 0.295 |
| 7 | Clean+LR+Noisy | Average filter | ✓ | ✓ | 0.456 | 0.360 | 0.338 |
| 8 | Clean+LR+Noisy | Denoise | ✓ | ✓ | 0.456 | 0.360 | **0.359** |

## 4.2 GATING NETWORK

We use softmax gating network. As we know the expert that is responsible for $X$, we consider the gating function as a classification problem. For the gating function $H$, the original images $x$ and the noisy images $F(x)$, the training objective is:

$$\mathcal{L}_{\text{Gate}}(H) = -\log(H(x)) - \log(1 - H(F(x))),\qquad(5)$$

where the output of $H$ is the result from the sigmoid function. We directly use the output of $H$ to select the input images for the object detection network among the candidate images from pre-processing experts.

## 5 IMPLEMENTATION

We use faster R-CNN (Ren et al., 2015) for object detection network and inception v2 (Szegedy et al., 2016) as a backbone feature extractor. Tensorflow object detection API (Huang et al., 2017) is used and modified to integrate MoPE model. We use MS-COCO 2014 dataset (Lin et al., 2014) for training and follow the default configuration except the learning rate schedule in Huang et al. (2017).

We use the up-sampled version of the 2x and 4x low-resolution images and noisy images along with the clean images for data augmentation. Gaussian noise $\mathcal{N}(\mu = 0, \sigma^2)$ is used where $\sigma$ is selected randomly in the interval [0, $max\_\sigma$] per each image. For every iteration, we randomly select one image between the clean and the low-resolution image. The selected image is injected with the corresponding Gaussian noise added image.

Faster R-CNN is trained with stochastic gradient descent (SGD) optimizer with momentum of 0.9 for 600k iterations. We start training with a learning rate of 2e-4 and divide it by 10 at 200k, 400k iterations. Denoise network and gating network is trained with Adam optimizer for 20k iterations. We also consider fine-tuning of faster R-CNN and MoPE model for extra 200k iterations with a learning rate of 2e-6.

## 6 EXPERIMENTAL RESULTS

We compare our approach against baseline network, network trained with data augmentation. We also compare average filter and denoise network for the pre-processing experts, and compare the network with/without gating network.

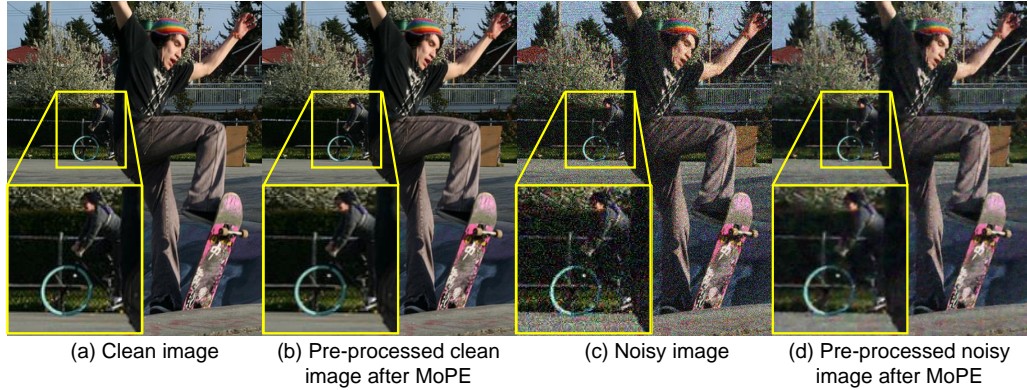

(a) Clean image    (b) Pre-processed clean image after MoPE    (c) Noisy image    (d) Pre-processed noisy image after MoPE

Figure 5: Sample images after the MoPE layer for the model 8 in Table 2

Table 3: Average tracking accuracy on MOT17 Train Set. Faster R-CNN Ren et al. (2015) with inception v2 Szegedy et al. (2016) backbone network

| | | MOTA | | |
|---|---|---|---|---|
| | Model 1 | Model 6 | Model 7 (Ours) | Model 8 (Ours) |
| Clean | 0.217 | 0.222 | 0.224 | **0.228** |
| $\sigma = 0.15$ | 0.161 | 0.170 | 0.191 | **0.197** |

## 6.1 OBJECT DETECTION

We first show the object detection accuracy on MS-COCO validation dataset as shown in Table 2.

**Effect of pre-processing:** The network trained only with the clean data shows reduced mAP for the noisy images (model 1). If we apply pre-processing like average filter or denoise network (model 2 and model 4), we observe increased mAP for the noisy images. Denoise network performs better than average filter at the cost of extra computation. For the clean images, however, we observe decreased mAP as the pre-processing degrades input feature information.

**Effect of gating network:** We apply gating network trained in isolation together with pre-processing unit. Model 3 and 5 in Table 2 show recovered mAP for the clean images as gating network guides the clean images not to be pre-processed. Eventually, the mAP for the clean images becomes similar with the baseline model (model 1).

**Effect of MoPE:** We compare our proposed methods with the network trained with data augmentation. We observe the network trained with various input images show improved mAP for the low resolution and noisy images (model 6) compared to the baseline network (model 1).

Next, we fine-tune the network from the baseline network (model 1) together with pre-trained gating network and denoise network. When fine-tuning, we don't use the loss function used in training the gating network and the denoise network. We only use classification and box regression losses used in object detection training. We apply the same data augmentation used for training model 6.

Our MoPE models (model 7 and 8) only improves mAP for the noisy images compare to the data augmented network (model 6). Gating network allows no-preprocessing for the clean and low-resolution images and injection of average filtered/denoised images during training further improves performance for the noisy images. We also observe denoise network performs better than average filter. Sample images after the MoPE layer for the model 8 are shown in Figure 5. As intended, the clean image is not affected by the denoise network and the noisy image is denoised with the denoise network.

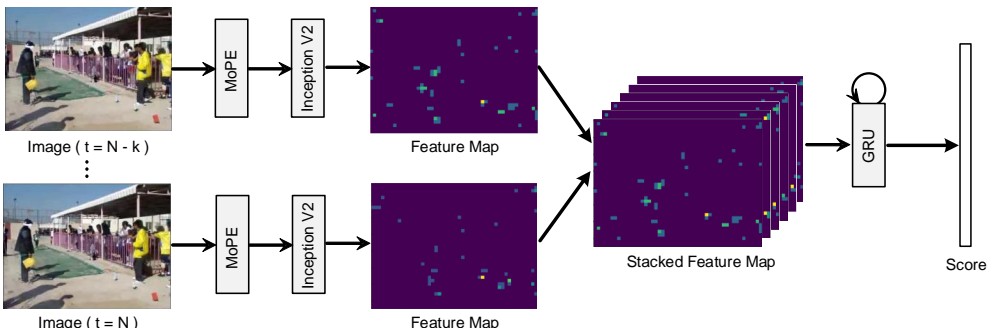

Figure 6: Activity classification network

Table 4: Human activity classification accuracy on UCF 101 Dataset. Faster R-CNN Ren et al. (2015) with inception v2 Szegedy et al. (2016) backbone network is used.

|              | Model 1 | Model 6 | Model 7 (Ours) | Model 8 (Ours) |
|--------------|---------|---------|----------------|----------------|
| Clean        | 0.511   | 0.509   | 0.522          | **0.531**      |
| $\sigma = 0.15$ | 0.157   | 0.461   | 0.492          | **0.492**      |

## 6.2 MULTIPLE OBJECT TRACKING

We adopt the approach implemented for object tracking presented by Bewley et al. (2016). It leverages the power of efficient object detector, uses Kalman filter (Kalman, 1960) for motion estimation and Hungarian method (Kuhn & Yaw, 1955) for data association. See Bewley et al. (2016) for more information.

Experimental evaluation was performed on the MOT-Challenge Dataset (Milan et al., 2016) which is primarily targeted towards multi-object tracking. We perform experiments on the train set which contains 7 sequences with varying lengths. 6 sequences have resolution 1920 x 1080 while one has a resolution of 640 x 480. It is a challenging dataset with multiple targets to track, occlusion effects, background clutter. 4 sequences are shot from a moving platform while the rest are filmed using static cameras.

We use all models as backbone detectors, and evaluate those for motion tracking problem for the clean and noisy videos. For reproducibility and consistency across our models, we provide the same seed before each sequence.

**Object tracking accuracy:** Our main metric of interest is multiple object tracking accuracy (MOTA) and the average bandwidth consumed for transmitting the sequences. The MOTA metric is a combined metric for evaluating three types of tracking errors i.e. false negatives, false positives and ID switches normalized by the number of ground truths within the sequence.

$$MOTA = (1 - \frac{\sum_i (fn_i + fp_i + id_i)}{\sum_i g_i}) \tag{6}$$

Average value of MOTA results are presented in Table 3. Network trained with data augmentation (model 6) shows better MOTA than the baseline (model 1). And our proposed method (model 7 and 8), especially for the model 8 with denoise network, shows better MOTA compared to other models for both clean and noisy condition. The results are consistent with the results in object detection, and this proves our proposed approaches are beneficial for both object detection and tracking.

## 6.3 HUMAN ACTIVITY CLASSIFICATION

We now apply our proposed approach on temporal network. Feature extractor of our object detection network is utilized for human activity classification. Experimental evaluation is performed on UCF

Table 5: Parameter size and the number of giga floating point operations (GFLOP) for the MoPE model. 244x244 is used for the input size. The numbers in the parenthesis represent the overhead compared to the object detection network with 300 proposals.

| | Denoise Network | Gating Network | Faster RCNN (inception V2) | |
| --- | --- | --- | --- | --- |
| | | | # of proposals = 20 | # of proposals = 300 |
| # params (MB) | 0.187 (0.45%) | 0.096 (0.22%) | 42.0 | 42.0 |
| Ops (GFLOP) | 0.171 (0.14%) | 0.034 (0.03%) | 9.47 | 116 |

101 Dataset which is an action classification/recognition dataset of realistic human action videos with 101 categories (Soomro et al., 2012).

Our activity classification network is shown in Figure 6. First, for each frame of images, MoPE determines if the image is clean or noisy and applies average filter and denoise network to noisy image. Then, backbone network extracts feature maps from "Mixed_4e" layer of inception v2 and these feature maps of six adjacent frames are stacked. One-layer Gated Recurrent Unit (Chung et al., 2014) with 500 hidden units classifies these stacked feature maps into one of human action categories.

The activity classification accuracy of four different configurations on UCF101 validation dataset is shown in Table 4. When Gaussian noise is added to evaluation data, classification accuracy of all configurations is reduced. Data augmentation during network training (model 6) shows much better classification accuracy on noisy data compared to the baseline network (model 1). MoPE enables classification accuracy on both clean and noisy image to be even better (model 7 and model 8). This experimental result shows that our proposed approach is also applicable on temporal network such as activity classification.

## 6.4 NETWORK COMPLEXITY ANALYSIS

We show the theoretical memory requirements and the number of floating point operations for our proposed method in Table 5. Even though actual execution time is platform dependent, those give an insight into the overhead of our proposed method. As shown in this table, the overhead of our method is very small in terms of both model size and the number of operations. As discussed in section 3.1, the nature of Gaussian noise can be detected with the minimal receptive field, the gating network architecture has been chosen to be small. Also, the denoise network is intentionally designed with the small number of filters. Even if we use smaller number of region proposals computed by the region proposal network, our MoPE has minimal effects on both memory and computation demands compared to the main object detection network.

## 7 CONCLUSION

We have proposed an efficient and noise robust object detection network for the resource constrained edge devices. To enhance the noise robustness with little overhead, simple yet efficient mixture of pre-processing experts (MoPE) model has been proposed. Adversarially trained auto-encoder network has been proposed as a denoise pre-processing expert for the noisy images. The overhead of adding gating network and denoise network is negligible considering heavy computation of object detection network. We have shown our proposed method can achieve better detection, tracking and human activity classification accuracy than the baseline network trained with data augmentation or the network attached with pre-processing modules.

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
