# OpenReview forum: "Mixture of Pre-processing Experts Model for Noise Robust Deep Learning on Resource Constrained Platforms"
_ICLR.cc/2019/Conference_

### Official Review · AnonReviewer1 · 2018-10-26
**A  reasonable approach for noise robust training but there is a lack of novelty**

**Rating:** 4
**Confidence:** 4

**Review:**

The paper addresses the problem of training an object detection network that can achieve good performance on both clean and noisy images.
The proposed approach is based on a gating network that decides whether
the image is clean or  noisy. in case of  noisy image a denoising  method is applied.  The network components form a mixture of experts architecture and are  jointly trained after a component-level pretraining.
How good is the gate performance? what happen if you use only one of the trained experts for all the clean/noisy  test data? It is not clear how you combined the results of the two experts. Are you computing a weighted average of the original and the enhanced images? Did you try to use a hard decision gating at test time?

---

> ### Author Response · Authors · 2018-11-26
> **Thanks for the feedback**
>
> Thank you for the valuable reviews.
>
> Q1 – How good is the gate performance?
> (Ans) The performance of the gating network was above 99% which haven’t included in the paper. Instead, we have shown sample images in figure 5 to show the effect of MoPE.
>
> Q2 - what happen if you use only one of the trained experts for all the clean/noisy  test data?
> (Ans) Please check the performance of the model 4 in table 2. Model 4 uses denoise network as a preprocessing for all the clean/noisy data without having gating network. The denoise net is trained on the noisy images with various noise levels including the clean images. When sigma is 0, the input image is actually the clean image (Please see the section 5 for implementation details).
>
> Q3 - It is not clear how you combined the results of the two experts.
> (Ans) Please see the figure 1. The shape of the preprocessing output is the same with that of the input image. Once we obtain the pre-processed images, the coefficient of the gating network is multiplied per each pre-processing and summed. The sum of the coefficients of the gating network is 1 as the output of the gating network is softmax function (See the section 4.2 for details).
>
> Q4 - Did you try to use a hard decision gating at test time?
> (Ans) No, we have used the same configuration as for the training time.

---

### Official Review · AnonReviewer2 · 2018-11-05
**Synthetic naive approach to handling distorted images by deep neural networks**

**Rating:** 4
**Confidence:** 5

**Review:**

The paper presents a synthetic naive approach to analyzing distorted, especially noisy, images through deep neural networks. It uses an existing gating network to discriminate between clean and noisy images, averaging and denoising the latter, so as to somewhat improve the results obtained if no such separation was used. It deals with a well known problem using the deep neural network formulation. Results should be compared to other image analysis methodologies, avoiding smoothing when not required, that can be used for the same purpose.  This should also be reflected in related work in section 2; the reason of including Table 1 in it seems unclear.

---

> ### Author Response · Authors · 2018-11-26
> **Thanks for the feedback**
>
> Thank you for the valuable reviews.
>
> Q1 – Results should be compared to other image analysis methodologies
> (Ans) The purpose of the paper is to enhance the performance of object detection and its related other tasks (multiple object tracking and activity recognition) under “both” noisy/clean condition with “limited overhead” in terms of memory/computation (table 5). Existing denoising techniques like BM3D [1] require significant computation per image, thus, are not practical to be used in real time embedded system. And use of gating network for avoiding smoothing when not required is not necessarily obvious. Gaussian noise is random, thus, one might think it is hard to learn pattern of gaussian noise using neural network which is considered to be good for the data having pattern. And we have showed that we can distinguish the clean and noisy images with gating network.
> Instead, we have added comparison between data augmentation techniques, with/without gating network and with/without fine-tuning to show the effect of the proposed MoPE in table 2/3/4.
> [1] K. Dabov, A. Foi, V. Katkovnik, and K. Egiazarian. Image denoising by sparse 3-d transform-domain collaborative filtering. IEEE Transactions on Image Processing, 16(8):2080–2095, 2007.
>
> Q2 - reason of including Table 1
> (Ans) We wanted to address that adding additional loss without changing network architecture doesn’t work for object detection which is not true for the image recognition. And this was the motivation of this work. We eventually proposed MoPE for object detection under clean/noisy/resolution variant conditions with small overhead (table 5).

---

### Official Review · AnonReviewer4 · 2018-11-13
**Weak novelty and significance**

**Rating:** 3
**Confidence:** 4

**Review:**

Summary
This paper introduced a parameterized image processing technique to improve a robustness of visual recognition systems against noisy input data. The proposed method is composed of two components; a denoising network that suppresses the noise signals in an image, and gating network that predicts whether to use the original input image or the one produced by the denoising network. The proposed idea is evaluated on three tasks of object detection, tracking and action recognition.

Originality and significance:
The originality of the paper is very limited since the paper simply combines the existing image denoising technique with the idea of gating. The practical significance of the work is also limited since the model is trained and evaluated with only synthetically generated noise patterns; it is not surprising that the proposed method (both denoising and gating networks) works under this setting, as the noise is created synthetically under the same setting in both training and testing. To demonstrate the practical usefulness, it would be great if the model is evaluated with the actual source of noises (e.g. noises from input sensors, distortion by image compression, etc).

Clarity:
I think the title of the paper is misleading; the proposed model is actually not a mixture of preprocessing units, as it combines *a* denoising unit together with identity mapping. The gating network is also not designed to incorporate a mixture of more than two preprocessing units, as it outputs only “on/off switches” instead of weights for K mixture components (K>2).

Minor comments:
1) the paper argued the importance of lightweight preprocessing but have not provided analysis on computation costs. From the current results, I don’t see the clear benefit of the proposed method (denoising network) over the average filtering considering the tradeoff between computation vs. performance.
2) In Figure 5, I suggest highlighting the differences among the examples for clarity.

---

> ### Author Response · Authors · 2018-11-26
> **Thanks for the feedback**
>
> Thank you for the valuable reviews.
>
> Q1 – Originality and significance:
>
> (Ans) In contrast to many other DL works focused on denoising or image classification on noisy images [1-2], the main contribution of this paper is to enhance the performance of object detection and its related other tasks (multiple object tracking and activity recognition) under “both” noisy/clean condition with “limited overhead” in terms of memory/computation (table 5). Also, we have discovered that adding average filter and U-net [3] like skip connection are beneficial for denoising.
> For the practical usefulness, it would be great if we can incorporate those actual noises as a future work. Thanks for your feedback.
>
> [1] P. Vincent, H. Larochelle, I. Lajoie, Y. Bengio, P.-A. Manzagol, "Stacked denoising autoencoders: learning useful representations in a deep network with a local denoising criterion", J. Mach. Learn. Res., vol. 11, no. 11, pp. 3371-3408, 2010.
> [2] S. Diamond, V. Sitzmann, S. Boyd, G. Wetzstein, and F. Heide. Dirty pixels: Optimizing image classification architectures for raw sensor data. arXiv preprint arXiv:1701.06487, 2017.
> [3] Olaf Ronneberger, Philipp Fischer, and Thomas Brox. U-net: Convolutional networks for biomedical image segmentation. In MICCAI (3), volume 9351 of Lecture Notes in Computer Science, pp. 234–241. Springer, 2015.
>
> Q2 – Clarity:
> (Ans) We think that the title of the paper is valid. It is true, in this paper, we have used identity mapping for the clean and low-resolution images as a preprocessing. This was the result from our experiments, not necessarily obvious one. There could be better preprocessing for low-resolution images that we haven’t explored.
>
> Q3 - lightweight preprocessing
> (Ans) Please see the table 5. Also, it is obvious that average filter requires less computation than the denoise net since denoise net includes average filter as a part (Please see the section 3.2 Pre-processing for the noisy images).
>
> Q4 - Figure 5:
> (Ans) We have changed the figure.

---

### Meta-Review · Area_Chair1 · 2018-12-04
**decision**

**Confidence:** 5
**Recommendation:** Reject

**Metareview:**

As the reviewers point out, the paper seems to be below the ICLR publication bar due to low novelty and limited significance.